# Fermented Wheat Bran Polysaccharides Improved Intestinal Health of Zebrafish in Terms of Intestinal Motility and Barrier Function

Qiuyan Chen [1,2,†], Jinju Mao [1,2,†], Yuan Wang [1,2,*], Na Yin [1,2], Na Liu [1,2], Yue Zheng [1,2], Xiaoping An [1,2], Jingwei Qi [1,2], Ruifang Wang [1,2] and Yanping Yang [1,2]

1   College of Animal Science, Inner Mongolia Agricultural University, Hohhot 010018, China
2   Inner Mongolia Herbivorous Livestock Feed Engineering Technology Research Center,
    Inner Mongolia Agricultural University, Hohhot 010018, China
*   Correspondence: wangyuan@imau.edu.cn
†   These authors contributed equally to this work.

**Abstract:** Intestinal barrier dysfunction and gut microbiota disorders have been associated with various intestinal and extraintestinal diseases. Fermented wheat bran polysaccharides (FWBP) are promising natural products for enhancing the growth performance and antioxidant function of zebrafish. The present study was conducted, in order to investigate the effects of FWBP on the intestinal motility and barrier function of zebrafish, which could provide evidence for the further potential of using FWBP as a functional food ingredient in the consideration of gut health. In Experiment 1, the normal or loperamide hydrochloride-induced constipation zebrafish larvae were treated with three concentrations of FWBP (10, 20, 40 μg/mL). In Experiment 2, 180 one month-old healthy zebrafish were randomly divided into three groups (six replicates/group and 10 zebrafish/tank) and fed with a basal diet, 0.05% FWBP, or 0.10% FWBP for eight weeks. The results showed that FWBP treatment for 6 h can reduce the fluorescence intensity and alleviate constipation, thereby promoting the gastrointestinal motility of zebrafish. When compared with control group, zebrafish fed diets containing FWBP showed an increased villus height ($p < 0.05$), an up-regulated mRNA expression of the tight junction protein 1α, muc2.1, muc5.1, matrix metalloproteinases 9 and defensin1 ($p < 0.05$), an increased abundance of the phylum Firmicutes ($p < 0.05$), and a decreased abundance of the phylum Proteobacteria, family Aeromonadaceae, and genus *Aeromonas* ($p < 0.05$). In addition, 0.05% FWBP supplementation up-regulated the intestinal mRNA expression of IL-10 and Occludin1 ($p < 0.05$), enhanced the Shannon and Chao1 indexes ($p < 0.05$), and increased the abundance of Bacteroidota and Actinobacteriota at the phylum level ($p < 0.05$). Additionally, 0.1% FWBP supplementation significantly improved the villus height to crypt depth ratio ($p < 0.05$) and increased the mRNA expression of IL-17 ($p < 0.05$). These findings reveal that FWBP can promote the intestinal motility and enhance the intestinal barrier function, thus improving the intestinal health of zebrafish.

**Keywords:** fermented wheat bran polysaccharides; intestinal motility; intestinal barrier; intestinal health; zebrafish



## 1. Introduction

As an important organ of the digestive and immune systems in human body, the intestine is in charge of nutrient digestion and the absorption of food [1,2]. In normal digestion, food is transited through the gastrointestinal tract by rhythmic contractions. Gastrointestinal dysmotility could lead to spasms or paralysis, thereby causing gastrointestinal diseases and morbidity [3]. Furthermore, intestinal constipation may change the gut microbiome, potentially contributing to the impairment of gastrointestinal functions. As the key determinant of gut health, the intestine is colonized by trillions of microbes

comprising the gut microbiota, which participates in nutrient metabolism, regulation of immune system and energy regulation [4]. Numerous observations have indicated that the changes in the composition of gut microbiota are associated with various intestinal diseases [5]. Therefore, the promotion of gastrointestinal motility and the stability of the gut microbiota are essential ecological characteristics, given their importance to gut health. Previous studies showed that various polysaccharides possess effects regarding the regulation of microbiota, enhancing immunity, and improving health status [6].

Wheat bran, as one available by-product of wheat processing, contains many high-value components: proteins, non-starch polysaccharides, enzymes and vitamins. In recent years, wheat bran polysaccharides (WBP) have attracted particular attention for their demonstrated beneficial effects, derived from their immunomodulatory, antioxidant, anti-hyperlipidemia, and antitumor activities [7–10]. However, wheat bran contains non-starch polysaccharides, which are not easily digested or absorbed by fish. Microbial fermentation technology can change the structure of non-starch polysaccharides. Recently, a novel polysaccharide component was isolated from fermented wheat bran in our laboratory. It is a 21.19 kDa hetero-polysaccharides (the total polysaccharide content was 96.96%), and it is mainly composed of glucose, xylose, arabinose, galactose and mannose [11]. Through our previous study, *S. cerevisiae* and *B. subtilis* were selected to fermented wheat bran, which could obtain the highest yield of fermented wheat bran polysaccharides (FWBP) [12]. In addition, our previous study showed that FWBP exhibited significant antioxidant activity in vitro, and exhibited stronger effects on growth performance and antioxidant function in juvenile zebrafish than WBP [13]. However, the influence of FWBP on gut health in zebrafish is poorly investigated.

Zebrafish are fresh water fish, and has been widely studied for various purposes, such as in ecotoxicology, immunity, and neurophysiology studies. When compared with murine models, which have high costs, long life cycles, and complex operation, zebrafish models' popularity is due in part to the fact that they have a shorter life cycle, easy and low-cost breeding, a high presence of human orthologous genes, and the availability of a large array [14]. In addition, the intestine of zebrafish is quite similar to that of humans [13,15,16]. In this aspect, zebrafish provide a useful platform for studying host–microbe interactions. Therefore, in the current study, we assessed the effectivity of FWBP on intestinal motility in zebrafish larvae. Furthermore, we sought to understand as to whether diets containing FWBP can alter the composition of the gut microbiota and modulate the gut immune response in zebrafish.

## 2. Materials and Methods

### 2.1. Preparation of Fermented Wheat Bran Polysaccharides

Wheat bran was fermented, and the FWBP was extracted from fermented wheat bran as described in our previous study [11]. Wheat bran was fermented according to previous methods, and the inoculum was prepared by mixing activated *S. cerevisiae* and *B. subtilis* in a ratio of 3.3:6.7, with a final concentration of $1 \times 10^8$ CFU/mL. The WB was inoculated with 10.4% (*v/v*) inoculum. Then, sterile, distilled water was added to achieve a 1:1.16 material: water ratio. The substrate was fermented at 36 °C for 47 h and dried at 45 °C for 48 h to obtain fermented WB. The fermented WB was ground and stored at 4 °C for the polysaccharide extraction.

### 2.2. Experiment 1: Effect of FWBP on Promoting Intestinal Peristalsis and Alleviating Constipation in Zebrafish Larvae

#### 2.2.1. Intestinal Peristalsis-Promoting Effect in Zebrafish Larvae

Zebrafish larvae at 5 dpf were randomly selected and placed in a 6-well microplate, with 3 mL 10 µg/L Nile red, for 16 h. Then, the zebrafish were washed with fresh embryo media to remove the dye, and FWBP was added at concentrations of 0 (Control), 10, 20, 40 and 30 µg/mL domperidone (DOM) for 6 h. At the end of the treatment, the zebrafish were rinsed with fresh embryo media, and we then randomly selected 20 zebrafish from each

concentration. The intestinal fluorescence intensity of the zebrafish was measured under a fluorescence microscope [3]. Then, according to the results of the intestinal fluorescence intensity of zebrafish, the promotion effects of DOM and FWBP were calculated as:

$$\text{promotion effect} = [(\text{DOM/FWBP group} - \text{Control group})/\text{Control group}] \times 100\%$$

### 2.2.2. Alleviating the Constipating Effect of FWBP in Zebrafish Larvae

Further study was carried out by establishing the constipation model of zebrafish. The constipation zebrafish model was established with a concentration of 10 µg/mL loperamide hydrochloride (LH). Then, we followed the steps outlined in Section 2.2.1. According to the results of the intestinal fluorescence intensity of zebrafish, the alleviating effects of DOM and FWBP were calculated as:

$$\text{alleviating effect} = [(\text{DOM/FWBP group} - \text{LH group})/\text{LH group}] \times 100\%$$

### 2.3. Experiment 2: Effects of FWBP on the Immune Activity, Intestinal Morphology and Gut MicroBiota of Zebrafish

#### 2.3.1. Animals and Experimental Diets

A total of 180 one-month old zebrafish (66.0 ± 0.7 mg) were randomly selected from 18 aquariums (3 L), at the rate of 10 fish per aquarium, and were adapted to a recirculating system for 14 days. After the nursery period, the zebrafish were weighed and divided into three treatments: (1) the basal diet (Control), (2) the basal diet supplemented with 0.05% FWBP (0.05% FWBP), and (3) the basal diet supplemented with 0.1% FWBP (0.1% FWBP). Six replicate tanks were randomly assigned per treatment group. The composition of the commercial diet was 38.9% crude protein, 15.1% crude fat, 93.6% dry matter, and 11% ash. Zebrafish were fed to apparent satiation for 8 weeks. In this study, the zebrafish were fed four times a day until apparent satiety. During the experimental period, water was exchanged automatically and the basic physicochemical parameters of the water, including the temperature, pH, and amount of dissolved oxygen, were maintained at 28 ± 1 °C, 7.2 ± 0.52, and 7.28 ± 0.39 mg/L, respectively.

#### 2.3.2. Histological of the Intestines

The intestinal samples were prepared for histological analyses, according to routine laboratory procedures. At the end of the feeding trial, segments of the middle intestine were surgically removed; three fish per tank (n = 18/group) were selected and preserved in freshly prepared 4% paraformaldehyde solution. Following fixation, the fixed intestine tissues were dehydrated in gradient ethanol, hyalinized in xylene, and embedded in wax. Embedded midguts were sectioned at 4–5 µm and stained with hematoxylin and eosin (HE), using a standard protocol. Then, we observed the sections using a microscope [17].

#### 2.3.3. RT-PCR Analysis

The total RNA of intestines from zebrafish was extracted using TRIzol Reagent (Invitrogen, Carlsbad, CA, USA), according to the manufacturer's instructions. To construct cDNA, the Super-Script III First-Strand Synthesis System (Invitrogen, Carlsbad, CA, USA) was used. The primers for gene expression detection are shown in Table 1. The real-time PCR method was used to determine the relative expression of genes, as described in our previous work [12]. β-actin was selected as the reference gene, and was used to normalize the gene expression levels. For each gene, the mRNA expression levels of the target genes were calculated using the $2^{-\Delta\Delta Ct}$ method, and data for each target transcript were normalized to the control zebrafish (1.0) [18].

**Table 1.** The primers for gene expression used in this study.

| Gene | Primer Sequences (5′→3′) | Product Size (bp) | Accession Number |
|---|---|---|---|
| IL-10 | F:ACGCTTCTTCTTTGCGACTG<br>R:TTGGGGTTGTGGAGTGCTT | 295 | NM_001020785.2 |
| IL-17 | F:CCCATCCATCCAATCAACAA<br>R:ACCTCAACGCCGTCTATCAG | 134 | NM_001020787.1 |
| TJPα | F:CCATTGAGACAGGAGTAAGCATT<br>R:ACATCACCAGAGGACTCAACAGA | 153 | XM_009303250.3 |
| Occludin1 | F:GATGTGGAGGACTGGGTCAATA<br>R:GCCGCTGCTAATAGGGACTG | 183 | NM_212832.2 |
| Mucin2.1 | F:GCCGCTGCTAATAGGGACTG<br>R:CGACAGTTTTCGATTTACGTG | 210 | XM_021470771.1 |
| Mucin5.1 | F:AATAATCTTGCCTGCCCAGAGT<br>R:CGACATTGATTTCAGTGATGTTCA | 190 | XM_021470622.1 |
| MMP9 | F:GCCTGCCAAATCAAGGAGTT<br>R:CGTTCACCATTGCCTGAGAT | 101 | NM_213123.1 |
| defensin1 | F:GCATCCTTTCCCTGGAGTT<br>R:AGCCTAATGGTCCGAAGTAAA | 91 | NM_001081553.1 |

IL-10: Interleukin 10; IL-17: Interleukin 17; TJPα: tight junction protein α; MMP9: Matrix metalloproteinases 9.

### 2.3.4. 16S rRNA Gene Sequencing Analyses of Zebrafish Gut Microbiota

Zebrafish were fed FWBP diets for eight weeks, then, four hours after the last feeding session, the digesta samples were collected from three biological replicates for each group [19], with six pooled samples per treatment as biological replicates. The gut microbiota from the experimental zebrafish was analyzed using bacterial 16S rRNA gene sequencing. Total genomic DNA from the intestinal content was extracted according to the manufacturer's instructions. Then, 16S rRNA genes from the genomic DNA samples were amplified by PCR using specific primers for the V4 region of bacterial 16S rRNA. The PCR amplification was performed with the following protocol: 30 s of initial denaturation at 98 °C; 98 °C for 10 s; 50 °C for 30 s; 72 °C for 30 s; repeated for 30 cycles; a final elongation step at 72 °C for 10 min. All procedures were conducted using the Novo-gene Bioinformatics Technology Co., Ltd. (Beijing, China). The sequencing library was quantified by Qubit and qPCR, and the barcoded V4 PCR amplicons were sequenced using a NovaSeq6000 platform, according to the manufacturer's protocol [20].

### 2.4. Statistical Analysis

All results for different treatment groups are shown as mean $\pm$ standard error (SE). The data were analyzed by one-way ANOVA in SAS. For treatments that showed significant differences, means were compared using the Tukey's test. The level of significance was set at $p < 0.05$.

### 3. Results

### 3.1. Experiment 1: Effect of FWBP on Promoting Intestinal Peristalsis and Alleviating Constipation in Zebrafish

FWBP was shown to promote intestinal peristalsis in the zebrafish, with the results shown in Figure 1. The intestinal fluorescence intensity of zebrafish in the morpholine group, as well as the FWBP groups with different concentrations, is significantly lower than that of the control group ($p < 0.05$). The intestinal motility in zebrafish, promoted by different concentrations of FWBP, increases and then decreases with increasing concentrations of FWBP. The intestinal motility promotion effect of FWBP in zebrafish at 20 µg/mL was 25.98%.

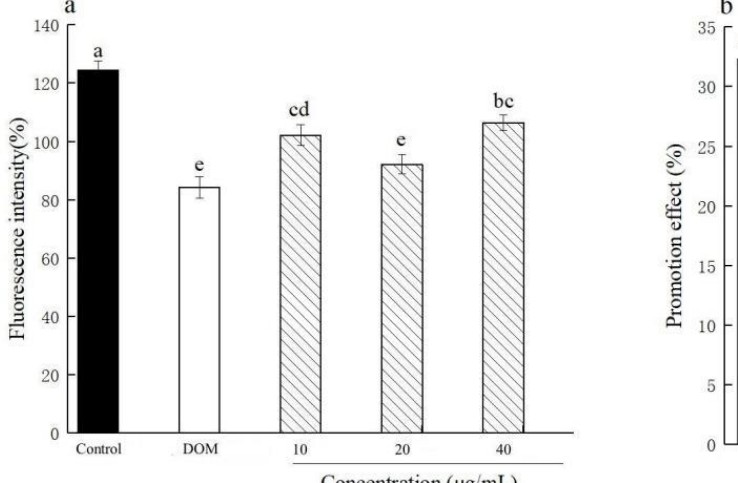
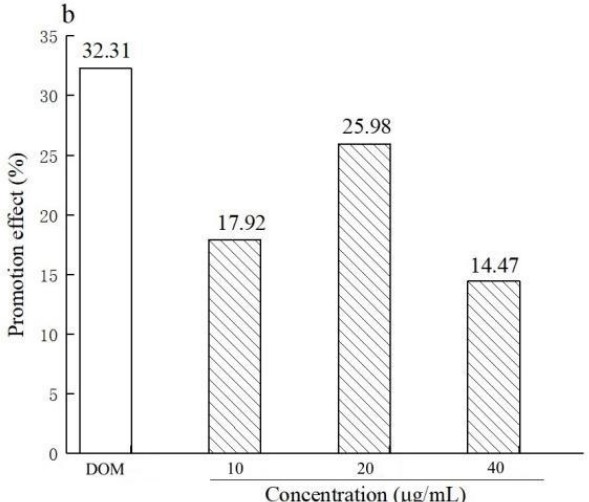

**Figure 1.** Effects of FWBP on intestinal peristalsis in zebrafish ($p < 0.05$). (**a**) Fluorescence intensity, %; (**b**) Intestinal motility promotion effect, %. DOM: Domperidone. Different lowercase superscript letters denote statistically significant differences among different groups ($p < 0.05$).

The effect of FWBP on the relief of constipation in zebrafish is shown in Figure 2. The fluorescence intensity of the LH group was significantly higher than that of the control group ($p < 0.05$), indicating that the constipation model was successfully established. However, zebrafish groups treated with morpholine, as well as 20 and 40 µg/mL FWBP, showed dramatically decreased fluorescence intensity. This result indicates that FWBP relieves intestinal constipation in zebrafish.

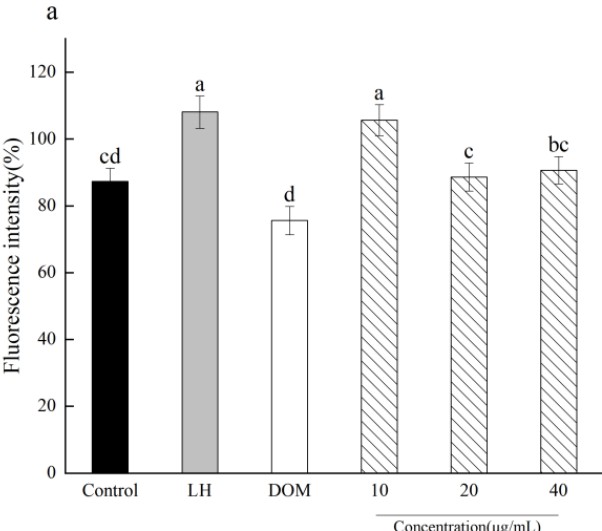
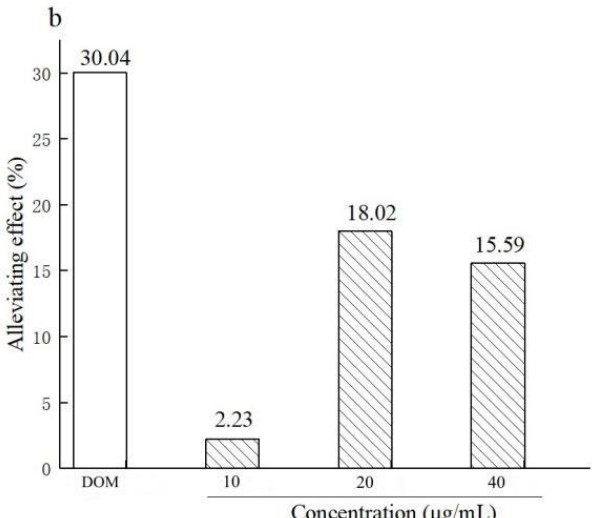

**Figure 2.** Alleviating effects of FWBP on intestinal motility inhibition in zebrafish ($p < 0.05$). (**a**) Fluorescence intensity, %; (**b**) Alleviating effect, %. LH: loperamide hydrochloride; DOM: Domperidone. Different lowercase superscript letters denote statistically significant differences among different groups ($p < 0.05$).

### 3.2. Experiment 2 Effects of FWBP on Immune Activity, Intestinal Morphology and Gut Microbiota of Zebrafish

### 3.2.1. Effect of FWBP on Intestinal Morphology

The effects of FWBP on the intestinal morphology of zebrafish are presented in Figures 3 and 4, and, compared with control group, the FWBP groups showed a significant improvement in the height of the villi ($p < 0.05$).

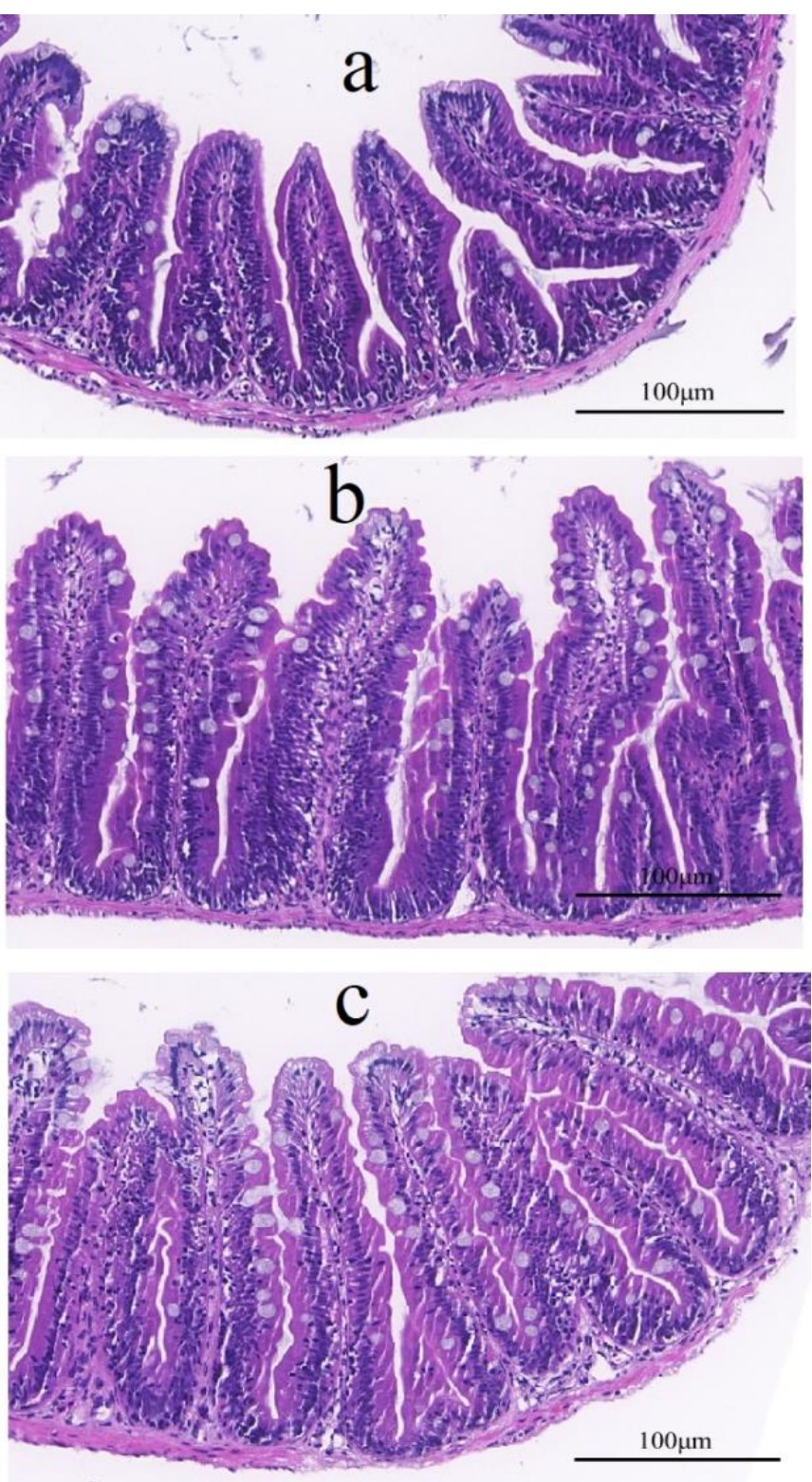

**Figure 3.** Representative HE-stained intestine sections from zebrafish fed with different diets ($p < 0.05$). (**a**) Control; (**b**) 0.05% FWBP; (**c**) 0.1% FWBP. FWBP: Fermented wheat bran polysaccharides. Scale bars = 100 μm.

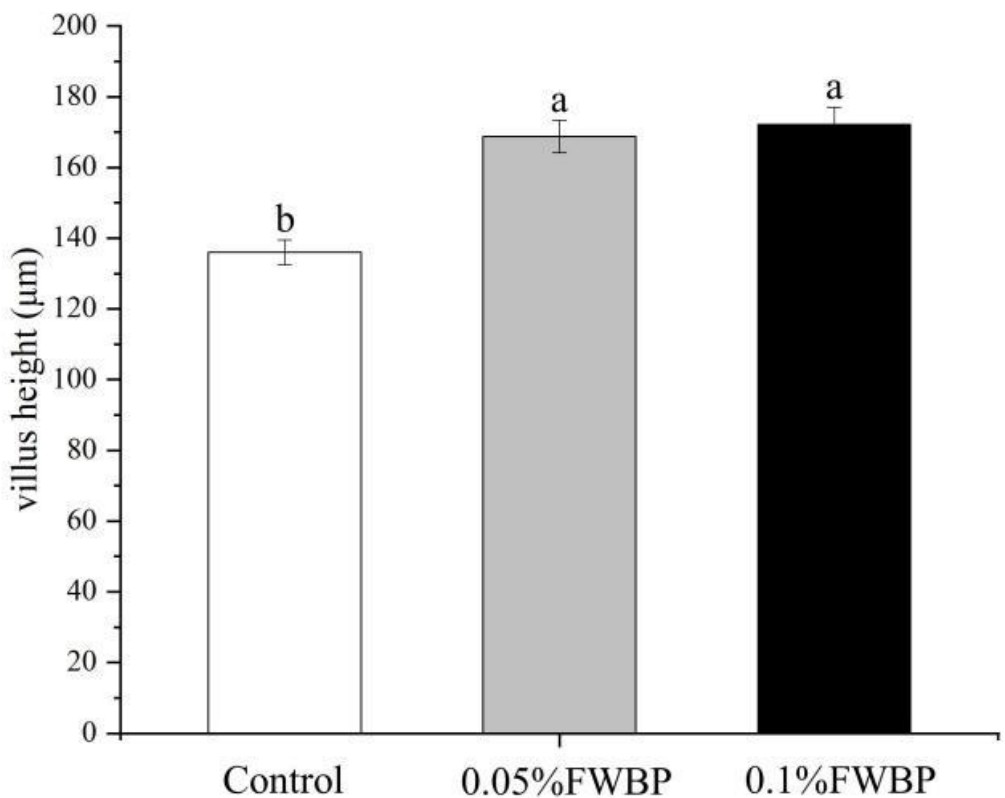

**Figure 4.** The effects of FWBP on villus height and crypt depth in the intestine. FWBP: Fermented wheat bran polysaccharides. Different lowercase superscript letters denote statistically significant differences among different groups ($p < 0.05$).

### 3.2.2. Expression of Intestinal Inflammation-Related Genes and Intestinal Mucosal Barrier-Related Genes

As shown in Figure 5, after zebrafish were fed diets with the supplementation of FWBP, the mRNA expression of interleukin 10 (IL-10) in the intestines of zebrafish in the 0.05% FWBP group was significantly increased, comparted to the control ($p < 0.05$). Zebrafish in the 0.1% FWBP group showed significantly increased IL-17 mRNA expression when compared with the control group ($p < 0.05$). The mRNA expression of intestinal mucosal barrier-related genes was determined, and the results are presented in Figure 6. When compared with the control group, FWBP groups showed up-regulated mRNA expression of tight junction protein $\alpha$(TJP1$\alpha$), muc2.1, muc5.1, matrix metalloproteinases 9 (MMP9), and defensin1 ($p < 0.05$). The 0.05% FWBP group showed higher Occludin1 mRNA expression than in the control and 0.1% FWBP groups ($p < 0.05$).

### 3.2.3. Effect of FWBP on Gut Microbiota
Diversity of Gut Microbiota

The Venn diagrams (Figure 7) show that a total number of 404 operational taxonomic units (OTUs) is shared by the three groups, and the numbers of unique OTUs in the control, 0.05% FWBP, and 0.1% FWBP groups were 214, 298, and 264, respectively. In the analysis of $\alpha$-diversity (Table 2), the Shannon and Chao1 indexes were significantly enhanced in the 0.05% FWBP supplementation group ($p < 0.05$).

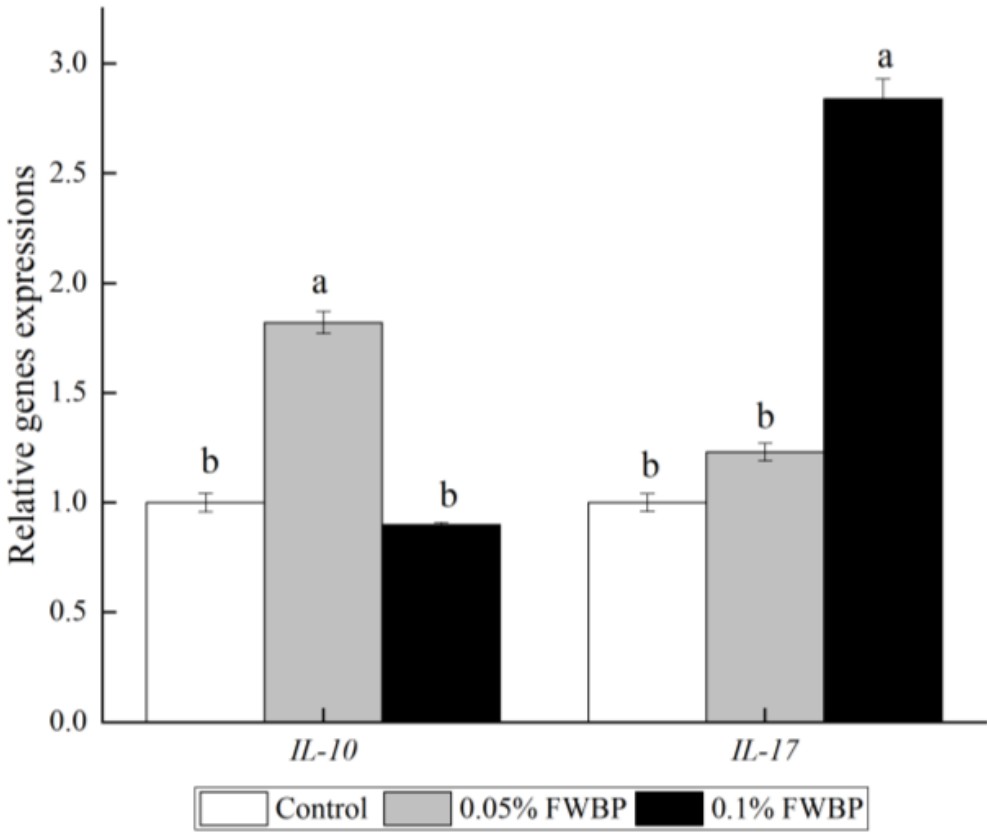

**Figure 5.** Gut inflammation-related gene expression in zebrafish fed with FWBP. FWBP: Fermented wheat bran polysaccharides. IL-10: Interleukin 10; IL-17: Interleukin 17. Different lowercase superscript letters denote statistically significant differences among different groups ($p < 0.05$).

Composition of Intestinal Microbiota

In regards to the intestinal microbial composition, at the phylum level (Figure 8), Fusobacteriota, Proteobacteria, Bacteroidota, Actinobacteriota, and Firmicutes were the predominant bacterial phyla in the gut of the zebrafish. At the family and genus level, Fusobacteriaceae and Aeromonadaceae, as well as *Cetobacterium* and *Aeromonas*, were the dominant families and genera, respectively (Figures 9 and 10). Taxonomic profiling showed a diverse gut microbiota community at the phylum, family and genus level. The significantly decreased abundance of Proteobacteria and the significantly increased abundance of Firmicutes at the phylum level were observed in 0.05% FWBP and 0.1% FWBP groups, in comparison with control group ($p < 0.05$). In addition, the 0.05% FWBP group showed a significantly increased relative abundance of Bacteroidota and Actinobacteriota ($p < 0.05$). At the family and genus level (Figures 9 and 10), by comparison with control group, a significantly decreased abundance of the family Aeromonadaceae and genus *Aeromonas* was observed in the 0.05% FWBP and 0.1% FWBP groups ($p < 0.05$).

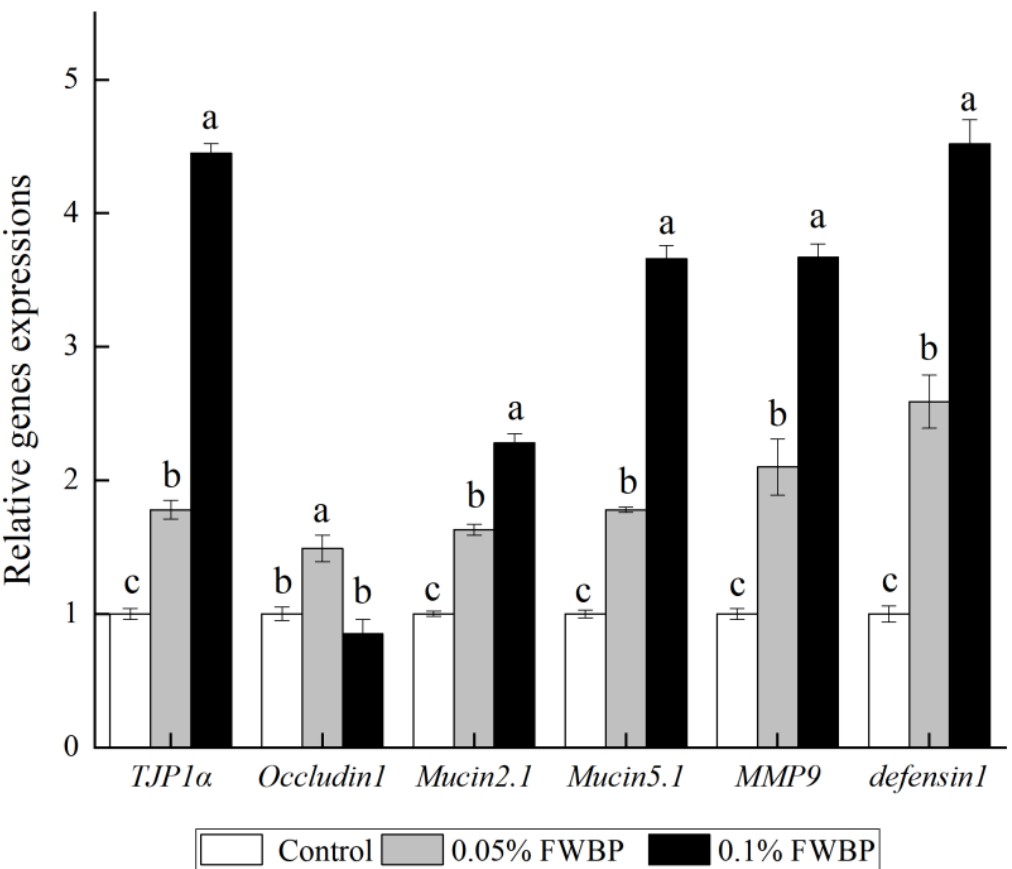

**Figure 6.** Gut mucosal barrier-related gene expression in zebrafish fed FWBP. FWBP: Fermented wheat bran polysaccharides. TJPα: Tight junction protein α; MMP9: Matrix metalloproteinases 9. Different lowercase superscript letters denote statistically significant differences among different groups ($p < 0.05$).

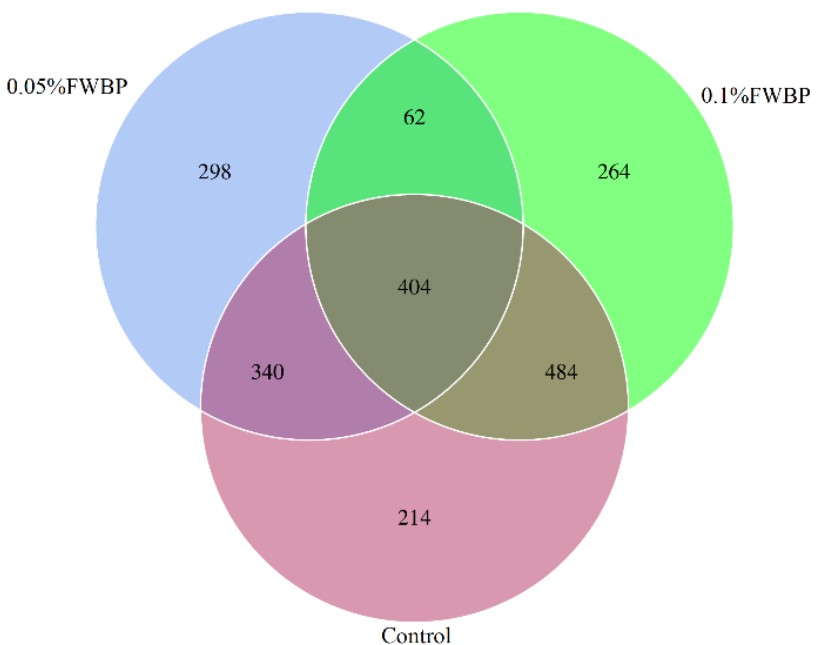

**Figure 7.** Venn diagram of OTUs among the dietary groups. FWBP: Fermented wheat bran polysaccharides.

**Table 2.** The effects of FWBP on α-diversity in the gut.

| Groups | Observed-Species | Goods-Coverage | PD-Whole-Tree | Shannon | Simpson | Chao1 |
|---|---|---|---|---|---|---|
| control | 341.67 ± 98.96 | 0.999 ± 0.0001 | 24.99 ± 4.02 | 1.76 ± 0.09 [b] | 0.61 ± 0.03 | 274.12 ± 74.44 [b] |
| 0.05% FWBP | 422.67 ± 78.65 | 0.999 ± 0.0001 | 41.93 ± 5.65 | 3.53 ± 0.51 [a] | 0.71 ± 0.09 | 565.43 ± 32.47 [a] |
| 0.1% FWBP | 342.66 ± 88.91 | 0.999 ± 0.0002 | 31.51 ± 6.29 | 1.63 ± 0.31 [b] | 0.57 ± 0.13 | 270.33 ± 46.10 [b] |

FWBP: Fermented wheat bran polysaccharides. Data are expressed as the mean ± SE. Different lowercase superscript letters denote statistically significant differences among different groups ($p < 0.05$).

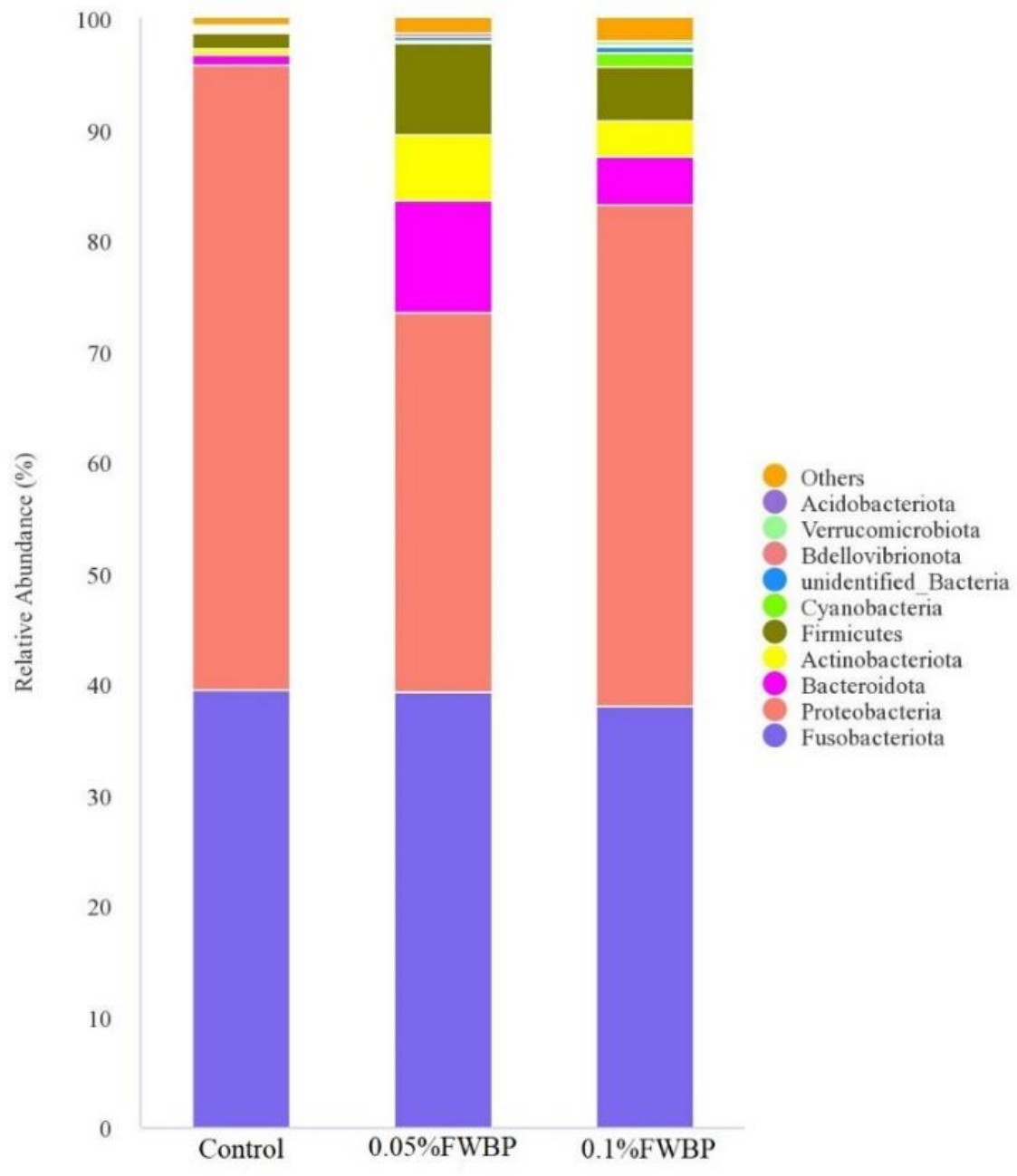

**Figure 8.** *Cont*.

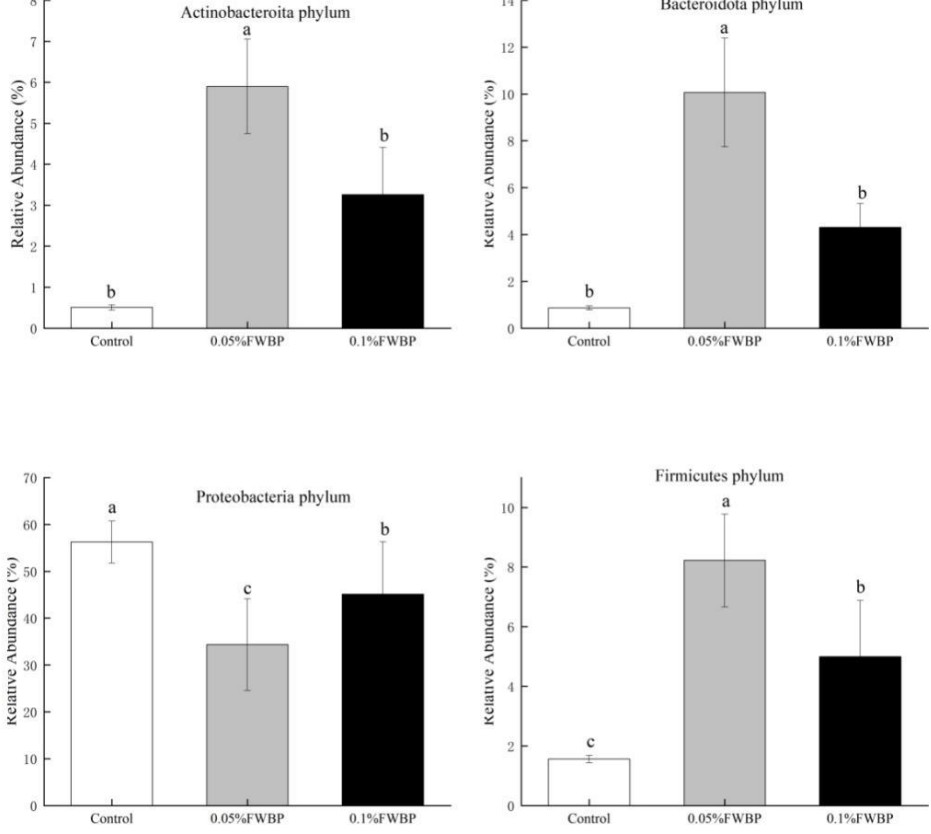

**Figure 8.** Microbial community in zebrafish fed with FWBP, at the phylum level. FWBP: Fermented wheat bran polysaccharides. Different lowercase superscript letters denote statistically significant differences among different groups ($p < 0.05$).

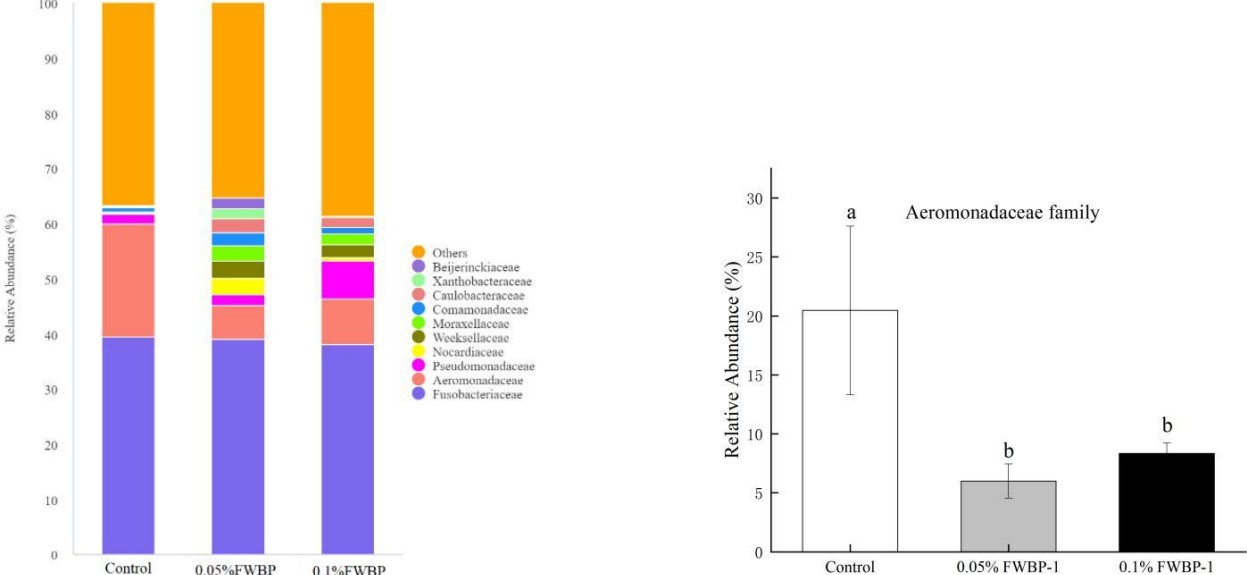

**Figure 9.** Microbial community in zebrafish fed with FWBP at the family level. FWBP: Fermented wheat bran polysaccharides. Different lowercase superscript letters denote statistically significant differences among different groups ($p < 0.05$).

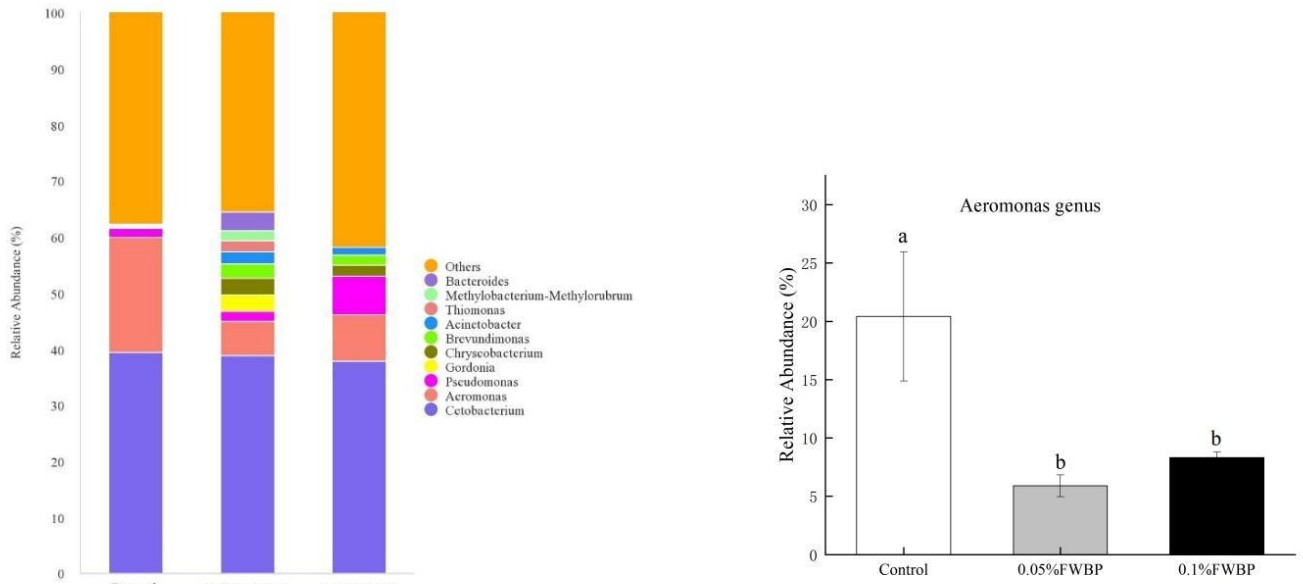

**Figure 10.** Microbial community in zebrafish fed with FWBP at the genus level. FWBP: Fermented wheat bran polysaccharides. Different lowercase superscript letters denote statistically significant differences among different groups ($p < 0.05$).

## 4. Discussion

Wheat bran, as a rich source of dietary fiber, contains many non-starch polysaccharides, including cellulose, and other non-cellulose polysaccharides, such as arabinoxylan, β- glucan, glucomannan, araban, and so on [21]. However, the gut tract of zebrafish cannot directly digest the non-starch polysaccharide. Therefore, the current study used microbial fermentation technology to break down wheat bran by microorganisms [22]. Studies have shown that wheat bran could be altered by microbial fermentation, and active substances, such as soluble polysaccharides, were significantly increased [23,24]. Based on our previous study, the results found that using *S. cerevisiae* and *B. subtilis* could change the structural characteristics of wheat bran and increase the yield of FWBP when compared to using a single strain. Because *S. cerevisiae* can secrete many enzymes and assimilate five-carbon sugars, *B. subtilis* can produce α-amylase, cellulase, β-glucanase, phytase, pectinase, xylanase, and so on [25]. In addition, our previous research found that the antioxidant activity of WBP can be improved by fermentation with *S. cerevisiae* and *B. subtilis,* both in vitro and in zebrafish models [11]. Previously, our study results showed that the FWBP group had better growth performance, higher antioxidant-associated gene expression, and a more positive effect on gut microbiota than the WBP group in zebrafish [13].

Intestinal motility disorders are an important cause of morbidity. The primary clinical manifestation of this effect is constipation, with the potential to produce intestinal obstruction, intestinal infarction, and a paralytic ileus, leading to death in sporadic cases [26]. The zebrafish is a potentially valuable model for gastrointestinal studies, due to its transparency, low cost, and ease of screening [27]. Studies have shown that loperamide hydrochloride can induce an increase in intestinal fluorescence intensity, which indicates that loperamide hydrochloride can be used to establish the zebrafish model of constipation [27]. Researchers have found that wheat bran polysaccharide induces cytokine expression via the toll-like receptor 4-mediated p38 MAPK signaling pathway, and prevents cyclophosphamide-induced immunosuppression in mice [7]. To determine whether FWBP can be used as a natural product for the treatment of intestinal motility disorders, we investigated FWBP's promotion of intestinal peristalsis in normal zebrafish and in zebrafish with loperamide hydrochloride-induced constipation. In the present study, the results showed that FWBP had a strong promoting effect on the intestinal motility of normal zebrafish. Furthermore, the fluorescence intensity was significantly altered in the zebrafish treated with loperamide

hydrochloride when compared with the normal group. After treatment with FWBP, the fluorescence intensity was dramatically decreased, indicating the amelioration of loperamide hydrochloride-induced constipation in zebrafish. This initial data suggested that the FWBP effectively promoted intestinal peristalsis.

Intestines with a large surface area acting as a barrier are acknowledged as the first line of resistance against external harmful substances' penetration into the intestine, associated with structural integrity, immunological status and microbiota homeostasis [28,29]. The villus height is a common indicator used to evaluate the function of the gut; an increase in villus height reflects an improvement in the digestion and absorption ability of the gut, and thereby positively affects the utilization of nutrients [2,30]. Additionally, complete intestinal barrier function is closely related to integrated intestinal morphology [31]. Polysaccharides have been shown to promote intestinal health in fish. However, there are very few studies regarding the effects of polysaccharides derived from wheat bran on intestinal function in zebrafish. In this work, the effect of FWBP on the histological structure of zebrafish gut was studied. Although the crypt depth did not show any alteration between experimental groups, villus height significantly increased in the FWBP-treated versus the control group. In addition, 0.1% FWBP significantly increased the villus height to crypt depth ratio. Our results were in partial accordance with Zahran et al., who observed that Nile tilapia fed with 1500 mg/kg *Astragalus* polysaccharides had an increased villus height in the anterior intestine [32].

Intestinal barrier function is considered to be the most important line of defense against external stimuli, which is composed of an immune barrier, mechanical barrier, chemical barrier, and biological barrier [33]. Inflammatory factors are a series of proteins secreted by endothelial cells, lymphocytes, monocytes, and fibroblasts that play an important role in regulating inflammatory processes, and are important facets of the intestinal immune barrier [34]. Our results displayed that the 0.05% FWBP group had anti-inflammatory effects on the intestine of zebrafish, via significantly increasing the expression of IL-10. It can be concluded that supplementation of appropriate amounts of FWBP in diets could improve the zebrafish intestinal immune system. In present study, the expression of the pro-inflammatory cytokine IL-17 was induced by the addition of 0.1% FWBP. However, the influence of FWBP on the intestinal immune system in zebrafish is poorly investigated, thus, it is difficult to make any direct comparison. Therefore, this aspect requires further investigation. In addition, our findings proposed that the intestinal mucosal barrier-related genes (TJP1$\alpha$, Occludin1, muc2.1, muc5.1, MMP9, and defensin1) exhibited higher gene expression in the FWBP-supplemented group than in the control groups, indicating that FWBP improved the zebrafish intestines' mechanical and chemical barrier function, by increasing the expression of related genes. This result may be due to the beneficial effects of FWBP on intestinal morphology. Similar results were also observed for other sources of natural polysaccharides in zebrafish diets. For instance, Li et al. reported that dietary supplementation with *Astragalus* polysaccharide up-regulated TJP1b, Occludin1 and IL-10 gene expression in the intestines of zebrafish [35].

The intestinal biological barrier is a mutually dependent and interrelated microecosystem, which is represented by the intestinal microbiota [33]. The intestinal microbiota plays important roles in immunity, the maintenance of homeostasis, and in digestion and nutrient absorption of the host [36]. A higher Shannon index value indicates a rich gut biodiversity. Our results demonstrate that supplementation with 0.05% FWBP can result in an enriched microbiota diversity in the zebrafish intestine. The possible explanation of the differences in results received from 0.05% FWBP and 0.1% FWBP could be that excess FWBP results in bacterial inhibition [37]. At the phylum level, the abundance of Proteobacteria was decreased, and the abundance of Firmicutes was increased in all FWBP groups. Proteobacteria is generally associated with dysbiosis, or an unstable gut microbial community; an increased abundance of Proteobacteria may bring potential risks to fish [38]. The Firmicutes are beneficial dominant bacteria in animals' guts, and many probiotics, such as *Lactobacillus*, *Enterococcus* and *Bacillus*, belong to the Firmicutes phylum [30]. In addition,

dietary addition of 0.05% FWBP remarkably enhanced the abundance of Bacteroidetes and Actinobacteria. Studies have shown that *Bacteroides* have the ability to produce SCFAs by fermenting dietary fiber and other undigested food remnants, in order to regulate host immune homeostasis [39]. As probiotics, bacteria in the phylum Actinobacteria can produce abundant secondary metabolites, which can inhibit the growth of pathogenic intestinal bacteria, and enhance the host defense ability [2]. Our present study found that dietary 0.05% FWBP had a stronger effect on probiotic levels. This was consistent with our previous in vitro study, which found that, with the increase of FWBP supplementation, FWBP had the effect of first increasing, then decreasing, probiotic levels [37]. This result may be that the high concentration of FWBP has an inhibitory effect on bacterial growth [30]. Previous studies of the effects of polysaccharides on aquaculture were consistent with our results. Su et al. confirmed that dietary Yu-Ping-Feng polysaccharide supplementation decreased the intestinal Proteobacteria and Chlamydiae abundance, and increased the abundance of Bacteroidetes in *Litopenaeus vannamei* [40]. At the family and genus level, the abundances of the family Aeromonadaceae and genus *Aeromonas* were greatly decreased in the all FWBP groups. *Aeromonas*, belonging to Aeromonadaceae, is mostly considered a major opportunistic pathogen in fish, causing intestinal inflammation. The results indicated that FWBP positively changed the gut microbiota through decreasing the abundance of opportunistic pathogens, and increasing the abundance of beneficial bacterium.

## 5. Conclusions

In conclusion, FWBP can promote intestinal peristalsis in zebrafish larvae. FWBP can improve intestinal morphology, mitigate intestinal inflammation, improve the mechanical and chemical barrier, and positively modulate the intestinal microbiota of zebrafish, thus enhancing intestinal barrier function. These results indicate that FWBP could be developed as a functional food ingredient candidate, in order to promote intestinal health.

**Author Contributions:** Conceptualization, X.A.; methodology, Q.C.; software, N.L.; validation, Y.W.; formala anlysis, Q.C.; investigation, Y.Y.; resources, R.W.; data curation, N.Y.; writing—original draft preparation, Q.C.; writing—review and editing, J.M. and Y.W.; supervision, Y.Z.; project administration, J.Q.; funding acquisition, J.Q., X.A. and Y.W. All authors have read and agreed to the published version of the manuscript.

**Funding:** This research was funded by Major Science and Technology Program of Inner Mongolia Autonomous Region, grant number 2021ZD0023-3, Major Science and Technology Program of Inner Mongolia Autonomous Region, grant number 2021SZD0024-4, Major Science and Technology Program of Inner Mongolia Autonomous Region, grant number 2020ZD0004, Key Technology Project of Inner Mongolia Autonomous Region, grant number 2020GG0030.

**Institutional Review Board Statement:** Not applicable.

**Informed Consent Statement:** Not applicable.

**Data Availability Statement:** Not applicable.

**Conflicts of Interest:** The authors declare no conflict of interest.

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
