# Peer review of "Fermented Wheat Bran Polysaccharides Improved Intestinal Health of Zebrafish in Terms of Intestinal Motility and Barrier Function"

_fermentation, doi:10.3390/fermentation9030293_

Round 1
Reviewer 1 Report (Previous Reviewer 1)
In my opinion, the Authors have responded adequately to the main criticisms posed by the Referees. However, several details (especially graphics) still need to be fixed. For example, between lines 78 and 79, the font size changes. I'm not sure if, "improved constipation" (lines 84 and 155), defines correctly the concept the authors meant, but I'm not a native English speaker. Why is statistical significance not indicated in panels b of Figures 1 and 2? In figure 3, I cannot see the scale bar. The Authors should indicate the statistical significance of the data relating to the depth of the crypts in table 2. In figures 4 and 5 (in particular), the Authors could use different colours (or greyscale) for the distinct columns. In line 287, a space after the word "barrier" is missed. I suggest authors check all graphic details throughout the text.
Author Response
Please see the attachment.

Reviewer 2 Report (New Reviewer)
The manuscript is very interesting and in general well conceptualized. However, even if the results of the digestion part are as expected, the journal is about Fermentation, therefore fermentation details must be strengthened:
P1) Please justify why pre-fermenting the fiber, if fiber is already a stimulant of the digestion and microbiota.
22) Please justify why do you use only S. cerevisiae and B. subtilis for the fermentation and not Lactobacillus.
3) Add the details of kinetics of fermentation.
4) The special issue is on antioxidant activity and the antioxidant activity must be added.
5) Please justify the lack of a control of non-fermented fiber.
Additional details needed for a better understanding of the manuscript:
66) Add a comparison of the advantages and disadvantages of Zebra fish model versus murine models.
77) Groups of 10 fish does not seem to be representative, please add comments and references on the validity of this small number.
88) Please avoid nonstandard abbreviations and explain (again) the abbreviatures in figures and tables.
Round 2
Reviewer 2 Report (New Reviewer)
The authors have improved the manuscript in accordance with the reviewers' suggestions and it can be published.
This manuscript is a resubmission of an earlier submission. The following is a list of the peer review reports and author responses from that submission.
Round 1
Reviewer 1 Report
The manuscript of Qiuyan Chen et al. describes the potential beneficial effect of a fermented wheat bran fiber on a loperamide-induced intestinal dysfunction model in zebrafish. The work would be interesting, but an experimental planning error, and a poor form of presentation, obliged me to ask for revisions.
Major revisions:
The potential healthy power of indigestible polysaccharides is well known. To convince the reader of the usefulness potential of fermented wheat bran fiber, the effects of FWBP, in comparison with other polysaccharides, must be described. The essential control is the description of the effects of unfermented wheat bran fiber in the same experimental model. Without this data, the manuscript cannot be accepted.
Minor but essential revisions concern the form of presentation:
- line 125: the Authors declare that they have proceeded following the methods of their previous works, but the citation reported is not theirs;
- lines 131-132: how the digested were collected. Authors should detail or, at least, cite a work;
- in lines 155-157: the Authors describe that: "The promotion of intestinal motility in zebrafish by different concentrations of FWBP increases and then decreases with increasing concentration". What could be the explanation for this phenomenon? Authors should formulate a reasonable hypothesis.
- Figure 2: what is "LH"? why is there no control in panel B?
- The title of table 2 appears incomplete, and the table lacks an acceptable legend. In addition, even if one can suppose the meaning of the letters in the table, it will be better specified. Furthermore, even if there are no significant differences in the third column, the Authors should indicate this with equal letters.
- In lines 179-180, the Authors assert: "However, no differences were observed in crypt depth of zebrafish intestine in all FWBP groups". Can the Authors give possible explanations for the observations?
- The legends of figures 4, and the following must be improved.
- The Figure between lines 223 and 224 has no title, caption or number. Consequently, the numbering of the figures mentioned in the text is wrong.
- lines 236-237: "Studies have shown that the loperamide can induce ..." a citation is necessary as well as an explanatory hypothesis of the observed phenomenons.
- lines 305-306: as already noted, the Authors need to insert essential controls!
Minor remarks:
- line 43: the digestives and immune are systems and not organs!
- lines 78-79: the Authors should give some details on fermentation techniques and not only a citation
- line 95: why did the Authors write: "micro-biota"?
- figure 3: the Authors have to put the microscopic magnification or, at least, the Authors should add a size bar in microns.
- the references have a double numbering.
Author Response
Dear Editor and honored reviewers,
Thank you for your contribution to reviewing work. This is the list of corrections of our manuscript entitled ‘Fermented wheat bran polysaccharides improved intestinal health of zebrafish in terms of intestinal motility and barrier function’ for publication in the Fermentation. For your suggestion we have dealt with the comments of the reviewers as follows:
Reviewer 1
The manuscript of Qiuyan Chen et al. describes the potential beneficial effect of a fermented wheat bran fiber on a loperamide-induced intestinal dysfunction model in zebrafish. The work would be interesting, but an experimental planning error, and a poor form of presentation, obliged me to ask for revisions.
Major revisions:
The potential healthy power of indigestible polysaccharides is well known. To convince the reader of the usefulness potential of fermented wheat bran fiber, the effects of FWBP, in comparison with other polysaccharides, must be described. The essential control is the description of the effects of unfermented wheat bran fiber in the same experimental model. Without this data, the manuscript cannot be accepted.
Response: Thanks for your comment. Our previous study showed that FWBP exhibited significant antioxidant activity in vitro and exhibited stronger effects on growth performance and antioxidant function to zebrafish juvenile than unfermented wheat bran polysaccharides. Therefore, this study selected fermented wheat bran polysaccharide as the research object to evaluate its effect on intestinal health of zebrafish. So far, no study has been done to evaluate the intestinal health effects of polysaccharides from wheat bran as feed additives in fish diets, therefore it is difficult to make any direct comparisons.
Minor but essential revisions concern the form of presentation:
- line 125: the Authors declare that they have proceeded following the methods of their previous works, but the citation reported is not theirs;
Response: Thanks for your comment. Our references have been added to the article.
- lines 131-132: how the digested were collected. Authors should detail or, at least, cite a work;
Response: Thanks for your comment. We have added references to how to collect digested.
- in lines 155-157: the Authors describe that: "The promotion of intestinal motility in zebrafish by different concentrations of FWBP increases and then decreases with increasing concentration". What could be the explanation for this phenomenon? Authors should formulate a reasonable hypothesis.
Response: Thanks for your comment. High concentrations may cause side effects, so the intestinal motility promotion decreases with increasing concentrations
- Figure 2: what is "LH"? why is there no control in panel B?
Response: Thanks for your comment. LH is short for loperamide hydrochloride.
- The title of table 2 appears incomplete, and the table lacks an acceptable legend. In addition, even if one can suppose the meaning of the letters in the table, it will be better specified. Furthermore, even if there are no significant differences in the third column, the Authors should indicate this with equal letters.
Response: Thanks for your comment. We have made changes in the article. Table 2 The effects of FWBP on villus height and crypt depth in the intestinal. FWBP: fermented wheat bran polysaccharides; abcp < 0.05 compared among different concentration. Scale bars = 100 µm.
- In lines 179-180, the Authors assert: "However, no differences were observed in crypt depth of zebrafish intestine in all FWBP groups". Can the Authors give possible explanations for the observations?
Response: Thanks for your comment. Crypt depth is closely associated with tissue turnover, in our study, FWBP showed no difference in the depth crypts of zebrafish intestinal, which may be because FWBP does not play a role in this aspect. Previous study showed that the increased villus height can improve the absorptive capacity of the intestine, and thereby positively affecting the utilization of nutrients.
- The legends of figures 4, and the following must be improved.
Response: Thanks for your comment. We have changed the legend of figure 4 to “Gut inflammatory-related genes and intestinal mucosal barrier-related genes expression of zebrafish fed with FWBP.”
- The Figure between lines 223 and 224 has no title, caption or number. Consequently, the numbering of the figures mentioned in the text is wrong.
Response: Thanks for your comment. We have captioned the figure between lines 223 and 224 and corrected the numbers mentioned in the text.
- lines 236-237: "Studies have shown that the loperamide can induce ..." a citation is necessary as well as an explanatory hypothesis of the observed phenomenons.
Response: Thanks for your comment. Our references have been added to the article. Loperamide has been used to induce constipation because it inhibits the secretion of water and the smooth movement of the intestinal wall, which causes long-term defecation and delays in intestinal transit, and loperamide-induced constipation animal models are widely used to identify therapeutic agents.
- lines 305-306: as already noted, the Authors need to insert essential controls!
Response: Thanks for your comment. We have inserted the essential controls.
Minor remarks:
- line 43: the digestives and immune are systems and not organs!
Response: Thanks for your comment. We have replaced ‘systems’ with ‘organs’ in line 43.
- lines 78-79: the Authors should give some details on fermentation techniques and not only a citation
Response: Thanks for your comment. Wheat bran was fermented according to previous methods, inoculum was prepared by mixing activated S. cerevisiae and B. subtilis in a ratio of 3.3:6.7, with a final concentration of 1×108 CFU/mL. The WB was inoculated with 10.4% (v/v) inoculum. Then, sterile distilled water was added to achieve a 1:1.16 material: water ratio. The substrate was fermented at 36 oC for 47 h and dried at 45 oC for 48 h to obtain fermented WB. The fermented WB were ground and stored at 4 oC for the polysaccharide extraction.
- line 95: why did the Authors write: "micro-biota"?
Response: Thanks for your comment. We have replaced ‘microbiota’ with ‘micro-biota’.
- figure 3: the Authors have to put the microscopic magnification or, at least, the Authors should add a size bar in microns.
Response: Thanks for your comment. The scale bars = 100 µm.
- the references have a double numbering.
Response: Thanks for your comment. We renumbered the references.

Reviewer 2 Report
The manuscript explored the effect of fermented wheat bran polysaccharides (FWBP) on intestinal health using zebrafish. However, the present form is not adequate. Comments are listed below.
1. The authors conducted two experiments but different units of concentrations are used (migrogram/mL and %). The readers would be confused.
2. The authors examined IL-10 and IL-17 mRNA expression. However, they need to show data concerning protein or cell levels.
3. They examined gut-mucosal barrier-related mRNA expression. Fig. 5 does not show the correct data. The results of mRNA expression are not adequate to prove the barrier function.
4. The IL-17 mRNA increased and diversity of gut microbiota did not enrich in 0.1% FWBP. These results seem to suggest that intestinal health and gut microbiota structure are not improved by 0.1% FWBP.
5. The gut microbiota composition was analyzed at phylum and genus levels. The discussion (L275-302) is vague. They need to analyze at family levels because 40% bacteria are not identified at the genus level.
Author Response
Dear Editor and honored reviewers,
Thank you for your contribution to reviewing work. This is the list of corrections of our manuscript entitled ‘Fermented wheat bran polysaccharides improved intestinal health of zebrafish in terms of intestinal motility and barrier function’ for publication in the Fermentation. For your suggestion we have dealt with the comments of the reviewers as follows:
Reviewer 2
The manuscript explored the effect of fermented wheat bran polysaccharides (FWBP) on intestinal health using zebrafish. However, the present form is not adequate. Comments are listed below.
- The authors conducted two experiments but different units of concentrations are used (migrogram/mL and %). The readers would be confused.
Response: Thanks for your comment. In experiment 1, fermented wheat bran polysaccharides were dissolved in distilled water and diluted to various concentrations (10, 20, 40 μg/mL). And the zebrafish treated with polysaccharide sample solution for 6 h. In experiment 2, fermented wheat bran polysaccharides were added to the basal diet (0.05% and 0.1%, w/w), feeding 8 weeks.
- The authors examined IL-10 and IL-17 mRNA expression. However, they need to show data concerning protein or cell levels.
Response: Thanks for your comment. This experiment is just a preliminary study, in our next experiments we will do a protein or cell levels analysis.
- They examined gut-mucosal barrier-related mRNA expression. Fig. 5 does not show the correct data. The results of mRNA expression are not adequate to prove the barrier function.
Response: Thanks for your comment. It is our negligence for the inconsistent data in Fig. 5 and in the manuscript. We have checked the data of Fig. 5 and remade the Fig. 5 with the correct data. Again, we are sorry for the mistake.
- The IL-17 mRNA increased and diversity of gut microbiota did not enrich in 0.1% FWBP. These results seem to suggest that intestinal health and gut microbiota structure are not improved by 0.1% FWBP.
Response: Thanks for your comment. Although the expression of IL-17 mRNA was increased and the diversity of gut microbiota was not increased in 0.1% FWBP, the barrier-related genes expression and abundance of Firmicutes at phylum was increased, abundance of Proteobacteria was decreased, which might be due to dose effect.
- The gut microbiota composition was analyzed at phylum and genus levels. The discussion (L275-302) is vague. They need to analyze at family levels because 40% bacteria are not identified at the genus level.
Response: Thanks for your comment. We have added family-level results to the text. At the family level, by comparison with control group, a significant decrease abundance of family Aeromonadaceae was observed in the 0.05%FWBP and 0.1%FWBP groups.

Round 2
Reviewer 1 Report
The authors have revised the text following all the indications of the referee. In the present form, the manuscript can be accepted for publication.
Reviewer 2 Report
The revised manuscript is not adequate. Comments are listed below.
Previous comment #2: The authors’ response “This experiment is just a preliminary study” does not make sense. I reckon that such a paper is not accepted in the international journal.
Authors’ Response: Thanks for your comment. This experiment is just a preliminary study, in our next experiments we will do a protein or cell levels analysis.
Previous comment #3: The results of mRNA expression are not adequate to prove the barrier function. They concluded “These findings revealed that FWBP can promoted the intestinal motility, and dietary FWBP enhance the intestinal barrier function and change the gut microbiota of zebrafish”. Thus, they should add data that prove the barrier function.
Authors’ Response: Thanks for your comment. It is our negligence for the inconsistent data in Fig. 5 and in the manuscript. We have checked the data of Fig. 5 and remade the Fig. 5 with the correct data. Again, we are sorry for the mistake.
Previous comment #4: The gut microbiota composition was analyzed at phylum and genus levels. As the discussion on the gut microbiota is vague, they should delete their discussion at phylum level. Instead of this, they discuss in details at family and genus levels. There are many differences that are interesting for readers.
1) Why did the diversity increased in 0.05% FWBP?
2) Why did 7 bacteria families and 8 bacteria genera increase in 0.05% FWBP?
3) Why did Psudomonus increase in in 0.05% FWBP?